# Initial Risk Assessment in Patients with Alveolar Echinococcosis—Results from a Retrospective Cohort Study

**DOI:** 10.3390/pathogens11050557

**Published:** 2022-05-09

**Authors:** Lynn Peters, Sanne Burkert, Jürgen Benjamin Hagemann, Rasmus Albes, Jonas Klemptner, Jessica Birkle, Elias Schwaibold, Sofia Siefermann, Beate Grüner

**Affiliations:** 1Department of Internal Medicine III, Division of Infectious Diseases, Ulm University Hospital, 89081 Ulm, Germany; sanne.burkert@uniklinik-ulm.de (S.B.); rasmus.albes@uni-ulm.de (R.A.); jonas.leif@uni-ulm.de (J.K.); jessica.birkle@uni-ulm.de (J.B.); elias.schwaibold@uni-ulm.de (E.S.); sofia.siefermann@uni-ulm.de (S.S.); beate.gruener@uniklinik-ulm.de (B.G.); 2Institute of Medical Microbiology and Hygiene, Ulm University Hospital, 89081 Ulm, Germany; benjamin.hagemann@uniklinik-ulm.de

**Keywords:** alveolar echinococcosis, risk factor, protective factor, progressive disease

## Abstract

**Background**: Alveolar echinococcosis (AE) is a potentially lethal parasitosis with a broad spectrum of disease dynamics in affected patients. To guide clinical management, we assessed initial prognostic factors for both progressive and controlled AE based on initial staging. **Methods**: A retrospective cohort study was conducted, examining 279 patients assigned to different clinical groups: cured, stable with and without the need for benzimidazole treatment, and progressive disease. Univariate analysis compared demographic and clinical variables. Significant variables were subsequently entered into two separate logistic regression models for progressive and controlled disease. **Results**: Based on the multivariate analysis, a large AE lesion (OR = 1.02 per millimetre in size; 95%CI 1.004–1.029), PNM staging (OR = 2.86; 95%CI 1.384–5.911) and especially the involvement of neighbouring organs (OR = 3.70; 95%CI 1.173–11.653) remained significant risk factors for progressive disease. A negative Em2+ IgG (OR = 0.25; 95%CI 0.072–0.835) and a small AE lesion (OR = 0.97; 95%CI 0.949–0.996) were significant protective factors. **Conclusions**: Patients with large lesions and advanced stages should be monitored closely and most likely require long-term treatment with benzimidazoles if curative resection is not feasible. Patients with small lesions and negative Em2+ IgG seem able to control the disease to a certain extent and a less strict treatment regimen might suffice.

## 1. Introduction

Alveolar echinococcosis (AE) is a potentially lethal parasitosis that extends throughout the northern hemisphere [1,2,3,4]. The adult tapeworms reside within the intestinal mucosa of carnivores—mainly foxes—as definite hosts, and produce eggs, which are excreted with the faeces and subsequently ingested by rodents as intermediate hosts. After ingestion, the larval stage of AE proliferates in the rodent’s liver and other organs. When definite hosts ingest infected tissue, an adult tapeworm will again develop within their bowel system, completing the life cycle. Humans are accidental intermediate hosts that are infected by ingesting viable eggs. Subsequently, the larval form of AE proliferates mainly in the liver like a malignancy, and can infiltrate neighbouring organs and metastasize [5]. In some patients, however, the larval lesion remains unchanged over decades. In both cases, patients remain asymptomatic for a long time [6,7]. Most patients are either identified incidentally or if symptoms occur at a late stage of the disease. Depending on the size and the localization of the larval lesion, patients can develop abdominal pain, jaundice, and cholangitis [8]. With the introduction of benzimidazoles (BMZ) as a parasitostatic treatment, mortality improved significantly from formerly more than 90% after ten years [5,9] to a life expectancy close to that of the general population [8,10,11]. In spite of this progress, courses of disease still vary greatly between individuals, ranging from asymptomatic patients without viable parasitic lesions to extensive disease involving multiple organs ultimately causing AE-associated death. The vast spectrum of clinical presentation and disease dynamics require personalized medicine and remain a challenge for clinicians [12]. Reliable tools are needed to predict the course of disease and guide clinical decision-making. This study aimed to assess initial, i.e., at the patient’s first presentation, risk and protective factors for progressive disease and controlled disease, respectively, to optimize patient-oriented care.

## 2. Results

### 2.1. Demographic and Descriptive Results

The descriptive characteristics are presented in Table 1. Mean follow-up time was similar between groups (cured: 62 months, stable: 46 months, and progressive: 45 months) and added up to 11,228 patient months with a median of 8 visits. Patients generally present at our centre every 6, 12, or 18 months. Most patients were diagnosed with either confirmed or probable AE. All patients except for *n* = 10 were started on BMZ treatment. The majority of patients were diagnosed with a high PNM stage, yet most had a stable disease during follow-up, and only a small fraction was progressive. In total, 2 patients died due to AE-associated complications. Since operated patients with ongoing treatment counted as stable, the number of cured patients is comparatively low. Two-third expressed some kind of symptoms before the diagnosis was established, mostly involving abdominal complaints.

### 2.2. Univariate Analysis

There was no difference in gender between the different outcome groups defined earlier. Regarding age, cured patients were younger (M = 42.5 years) compared to stable or progressive patients (M = 54.9 years) (F (3) = 9.319, *p* < 0.001). There was no significant difference in age between patients with controlled and progressive disease. In total, 33 of 190 patients (17.4%) had a recognised occupational disease. One-third of the patients with progressive AE had an occupational disease, while the percentage within the other groups was non-significantly smaller. Co-morbidities such as immunosuppression or malignancies did not influence the outcome significantly.

At first presentation, patients with progressive disease reported significantly more AE-related symptoms than patients with controlled disease, most likely due to the advanced involvement of affected organs (Χ^2^(1) = 4.416; *p* = 0.036). Staging according to the PNM classification differed significantly between outcome groups (Χ^2^(3) = 35.536, *p* < 0.001). Of those with progressive disease, 82% were staged as P4, which was only the case in 51% of those with stable disease with BMZ treatment, 24% in cured patients, and 22% in those with stable disease without BMZ treatment. Similarly, in 68% of patients with progressive disease neighbouring organs were involved, which applied to only 38% of stable patients with, and 21% of stable patients without BMZ treatment and 25% of cured patients (Χ^2^(3) = 15.487, *p* = 0.001). There was no significant difference between groups regarding distant metastases. Based on the respective PNM status, staging differed significantly between the clinical groups (c.f. Figure 1) (Χ^2^(12) = 56.313, *p* < 0.001).

Regarding laboratory results at first presentation, mean activity of ALT and levels of γGT, AP, bilirubin, and CRP were highest in patients with progressive disease and lowest in cured patients. Patients with stable disease showed intermediate levels for all biomarkers. However, the differences were not significant (c.f. Appendix A). Only differences in γGT levels tended towards significance (t (35) = −1.916, *p* = 0.064), with progressive patients presenting a mean level of 184 U/l and stable patients without the need for BMZ treatment 113 U/l. The level of γGT correlated with the occurrence of biliary complications (s = 0.524, *p* < 0.001).

Group differences regarding the levels of IgE and serological markers at initial presentation were impressive and highly significant (c.f. Figure 2 and Appendix A). The highest values were found in patients with progressive disease and the lowest in those with stable disease without need for BMZ. On average, the latter exhibit normal values of Ech. IgG IHA and a negative Em2+ serology in 58% of cases. With a cut-off at 100 IU/mL for IgE levels and 15 U/l for Ech. IgG EIA, the levels reached are barely above the threshold.

Regarding the size of the parasitic lesion at first presentation, patients with progressive disease presented with comparatively large lesions, while those with stable disease without need for BMZ had the smallest lesions (c.f. Figure 3A) (F(3) = 11.614; *p* < 0.001). While AE lesions in patients with progressive disease extended to 107 mm on average, lesions in stable patients with and without the need for BMZ were significantly smaller, with 64 mm and 38 mm, respectively. The morphology according to the EMUC-US classification [15] differed between clinical groups (Χ^2^(21) = 148.512; *p* < 0.001): a large proportion of patients with stable disease without need for BMZ presented with metastasis-like lesions, which are usually small lesions. In this group, there were no cases with a pseudocystic pattern. On the other hand, high numbers of pseudocystic lesions were observed in patients with progressive disease, whereas in this group, there were no cases with metastasis-like or ossification patterns (c.f. Figure 3B).

Correlation between the respective serological markers, levels of IgE, and size of the largest AE lesion revealed significant or highly significant results (c.f. Table 2).

The number of unplanned treatment interruptions was higher in patients with progressive disease (M = 36%) than in other groups (M = 9–21%) (Χ^2^(3) = 12.481; *p* = 0.006). Cured patients exhibited a higher albendazole blood level (M = 0.70 mg/L) than other groups (M = 0.37–0.48 mg/L) (F (3) = 3.082, *p* < 0.001).

Further results for all analysed variables, including statistical tests and findings for each group, are displayed in the Appendix A.

### 2.3. Multivariate Analysis

Two regression models were established to analyse (a) risk factors for progressive disease and (b) protective factors for controlled disease, adjusting for possible confounders and effect modifiers (c.f. Table 3). ‘Controlled disease’ is the subgroup of those stable without the need for BMZ without prior resection. Thus, the parasite is controlled by the host’s immune defence. A negative Em2+ serology increases the odds of controlled disease by 75% compared to a positive serology. Regarding progressive disease, the initial involvement of neighbouring organs increased the odds of progression 3.7-fold. Furthermore, the disease stage was a significant risk factor for progressive disease, as with every increase in stage, the risk for disease progression increased 2.9-fold. The size of disease was prognostic for both controlled and progressive disease: with every millimetre the AE lesion increased or decreased, the odds of a progressive course of disease increased by 2% or decreased by 3%, respectively. If AE size was excluded from the model for progressive disease, ‘P’ became close to significant (*p* = 0.054).

## 3. Discussion

Since benzimidazoles have been introduced for the treatment of AE in the 1980s [16], mortality and morbidity due to progressive courses and complications have been successfully reduced [5,8,10,17,18]. Even though the recommendations for medical treatment remain unaltered [13,19], this trend is still ongoing. The number of cured patients, i.e., those who underwent resection without signs of residual or regressive disease, increased from 5–7% before 2000 [8,20] to 14% in our cohort, a trend observed similarly in other cohorts [10]. The proportion of patients with progressive disease decreased from 10.5% [20] to 5% in our cohort. Hence, the proportion of those with stable disease increased from 55% to 62% [20]. This shift might be attributed to a higher proportion of early-stage AE (stage I or II) in our cohort compared to earlier cohorts [8]. In contrast to late-stage AE, early lesions allow for curative resection or remain stable with early medical treatment [8,21]. The increased availability and accessibility of imaging techniques at all healthcare levels might explain this change over time. Early-stage AE is often an incidental finding, meaning that symptoms such as abdominal discomfort and pain have not yet occurred [5,8,21].

Early diagnosis and treatment of AE is therefore important in reducing morbidity and mortality. Nevertheless, clinical courses and outcomes still vary greatly between individual patients. Certain red flags should be evaluated at initial presentation and guide the clinical management.

First, the size, stage, and morphology of the AE lesion differed between outcome groups. An increasing or decreasing diameter proved to significantly raise or lower the odds for an adverse outcome. Prior research indicated that low PNM stages tended towards a beneficial outcome [8], and that advanced stages were associated with increased mortality [21]. Our results confirm that staging according to the PNM classification at first presentation is an important and reliable tool to predict the clinical course of disease, especially ‘N’, the involvement of neighbouring organs. In terms of ‘P’, lesions close to the liver hilus have been considered a risk factor [5,17] since they are more likely to cause biliary or vascular complications, explaining the high proportion of P4 lesions in patients with progressive disease in our cohort, even though ‘P’ did nor remain significant in the multivariate model. In contrast to earlier results, group differences concerning the occurrence of distant metastases were not significant in our patient population [8,17]. Regarding AE morphology according to EMUC-US [15], metastasis-like lesions were the most common pattern in patients with stable disease without need for BMZ, yet never occurred in patients with progressive disease. On the other hand, lesions with a pseudocystic pattern were mainly found in patients with progression, yet never in those stable without the need for BMZ. AE morphology, however, lost its significance when entered into the multivariate analysis. Therefore, reasons might be that: (1) lesions with a metastasis-like pattern are the smallest and those with a pseudocystic pattern are the largest lesions [22], and morphology acts merely as a confounder; and (2) morphology was only described in 16 patients with progression, limiting the statistical power.

Moreover, our data demonstrate that the levels of serological markers (IgG IHA, IgG EIA, Em2+) and IgE differed significantly between outcome groups at initial presentation and were hence predictive for the later course of disease. Patients who developed a progressive disease during follow-up already stood out with higher levels of respective markers at first presentation compared to those with stable disease. Within the latter group, those without need for BMZ presented with lower markers than those who required BMZ. Additionally, the level of respective makers correlated with the diameter of the lesion. Thus, we conclude that IgE and serological markers at initial presentation are surrogate markers for disease activity and size, which in turn influence the course of disease. Vice versa, low levels of the respective markers suggest immunologically ‘cold lesions’, i.e., controlled disease (not to be confused with abortive or died-out lesions, which are calcified or in which viability was excluded histologically [6,7]). The use of serological markers to guide clinical decisions, e.g., to end medical treatment, is common practice at different reference centres [3,10,23,24,25,26]. Our data confirms that a negative serology is an important condition before an attempt to interrupt medical treatment should be made. The close correlation with size might explain why some markers failed to reach significance in the multivariate analysis, except for a negative Em2+ which increased the odds for controlled disease by 75%.

Finally, unplanned treatment interruptions had occurred significantly more often in patients with progressive disease, which has also been shown earlier [8]. Other studies indicate a higher rate of recurrence after surgery if BMZ were not administered concomitantly [10]. Thus, conservative therapy with BMZ remains invaluable in the treatment of AE. Mean ABZ levels were higher in cured patients, who receive a shorter treatment regimen compared to stable patients with long-term BMZ intake who—based on the authors’ experience—often wish a decrease in dosage after several years. There was no difference in ABZ level between patients with progressive or controlled disease, indicating that bioavailability in both groups should be equal.

Interestingly, in our cohort, neither immunosuppression nor malignancies were associated with progressive disease. These results are in line with previous research: Chauchet et al. (2014) [27] proved an overall beneficent outcome in immunosuppressed patients with AE, and Lachenmeyer et al. (2019) [28] found no influence of immunosuppression on disease recurrence in the multivariate analysis.

## 4. Methods

### 4.1. Sample Size Calculation and Data Collection

We conducted a retrospective cohort study including all patients initially presenting at the University Hospital of Ulm between 01/2011 and 12/2018 with alveolar echinococcosis (*n* = 279). The study was approved by the Ethics Committee of the University of Ulm (protocol code no. 420/20). Following the descriptive analysis, 6 patients with progressive disease who presented at our centre before 2011 were added before the risk factor analysis to ensure a statistical power >80%. Patients present at our centre every 3 to 18 months, receiving a clinical examination, laboratory testing, and imaging. Data were extracted from the patients’ electronic records and pseudonymised before further handling.

### 4.2. Variables of Interest

#### 4.2.1. Demographic Variables, Medical History and Clinical Presentation

We assessed age, gender, and if AE was classified as an occupational disease, meaning that AE was acquired in context with farming activities. After assessment of AE-related symptoms, patients were classified as ‘asymptomatic’ if no symptoms were reported and ‘symptomatic’ if one or more symptoms were reported. Furthermore, we documented if patients had received immunosuppressive co-medication equivalent to ≥20 mg prednisolone for at least 2 weeks and if malignant co-morbidities were present, since malignancies can also compromise the immune defence. We included immunosuppression and malignancies that occurred both before and after diagnosis of AE.

#### 4.2.2. Laboratory Examinations

Levels of alanine transaminase (ALT), gamma-glutamyltransferase (γGT), alkaline phosphatase (AP), bilirubin, and C-reactive protein (CRP) at first presentation were recorded. Furthermore, we assessed levels of immunoglobulin E (IgE) and serological markers. The latter included Echinococcus IgG IHA (Cellognost Echinococcosis, Dade Behring, Germany), Echinococcus IgG EIA (VIRION/SERION ELISA *classic* Echinococcus IgG ESR107G, Würzburg, Germany), and IgG antibodies directed against Em2+ (Bordier Affinity, Crissier, Switzerland).

#### 4.2.3. Imaging, Staging and AE-Related Complications

The largest diameter of the largest AE lesion was documented at every visit using ultrasound (US), computed tomography (CT), magnetic resonance imaging (MRI), or positron emission tomography with 2-deoxy-2-[fluorine-18]fluoro-D-glucose and computed tomography (PET-CT). The morphological classification of AE lesions according to EMUC-US was introduced at our centre during the study period [15]. Morphology descriptions from the last visit were evaluated and are missing in operated patients.

All AE infections were classified according to the PNM classification [14]. In short, the location of the parasitic lesion ‘P’ rates from 0 to 4, a higher number indicates an adverse hepatic location involving critical structures, while ‘N’ indicates the involvement of neighbouring organs and ‘M’ distant metastases. The PNM classification translates to the different disease stages I, II, IIIa, IIIb, and IV [14].

Occurring complications, such as cholestasis, cholangitis, portal vein/ vena cava involvement, or portal hypertension, were diagnosed with either US, CT, MRI, or PET-CT. Complications were considered AE-related if a causal relation was proven or highly likely. Cholestasis was defined as dilatation of the *ductus hepaticus communis* in US/CT/PET-CT or MRI and simultaneous elevation of serum bilirubin.

#### 4.2.4. Treatment and Outcome

We documented if at least one unplanned treatment interruption occurred during the course of disease, e.g., due to intolerance or incompliance. Moreover, the albendazole blood level of every visit was recorded and the ‘mean albendazole blood level’ across all visits was calculated.

Patients were assigned to different clinical groups according to the following outcomes (c.f. Table 4).

In order to achieve the most clear-cut results, we focused our comparative analysis on patients with controlled and progressive disease, as these subgroups present the extreme ends of the disease spectrum, and are thus the most interesting from an immunological perspective.

### 4.3. Statistical Analysis

Descriptive results are presented with mean (M), median (MD), and standard deviation (SD). All variables were tested for normal distribution (Kolmogorov-Smirnov test or Shapiro-Wilk test). Prior to the univariate analysis assessing the difference between clinical groups, patients with only one visit were excluded. Differences between groups were analysed using ANOVA and *t*-test for the normally distributed, or Kruskal-Wallis test and Mann-Whitney U test for non-parametric data and ordinally scaled variables. For categorical variables, the Chi-square test was applied. Correlations were performed according to Pearson (continuous variables) or Spearman (non-continuous variables). To evaluate risk and protective factors for progressive and controlled disease, respectively, a logistic regression analysis was conducted, adjusting for confounders and interactions between variables. Only variables that differed significantly between the subgroups of interest in the univariate analysis were entered into the model to prevent over-fitting. Before the final model was established, a screening for multicollinearity was conducted. The Ech. IgG IHA was excluded from the multivariate analysis because of multicollinearity, whereas Ech. IgG EIA, Em2+, IgE, and the size of the lesion were not multicollinear (VIF < 3) despite close correlation and hence remained in the model. Due to multicollinearity, either ‘P’ and ‘N’ or the stage of disease were included. Prior to the multivariate analysis, multiple imputation was performed to adjust for missing variables. As many auxiliary variables and cases as possible were added to the imputation process to produce an accurate dataset. The number of data sets generated was determined using the formula suggested by Newgard and Haukoos [29], aiming for a relative efficiency of at least 95%, which is considered a high rate. Results were considered significant with a *p*-value smaller than 5%. Statistics were calculated using IBM SPSS V25 and graphs were created with Microsoft Excel 365 VS 2112.

## 5. Conclusions

Overall, our data suggest that patients with large lesions or extensive disease according to the PNM classification, as well as those with strongly elevated serological markers and IgE, should be monitored rather closely and most likely require long-term continuous BMZ treatment if curative resection is not possible. On the other hand, small lesions, e.g., with a metastasis-like pattern and a negative Em2+, might indicate that a patient’s immune system can control the disease to a certain extent and might need BMZ treatment, if at all, only for a short period or with a lower dosage. Therefore, these patients might not necessarily benefit from surgical intervention.

## 6. Limitations

Since AE is usually a slow growing disease, the follow-up period of this study might be too short to assess long-term outcomes. Especially in those considered to control the disease, a reactivation and hence growth of the AE lesion in the future cannot be ruled out at this point in time. Therefore, a future re-assessment of this subgroup seems appropriate. Furthermore, the retrospective and monocentric study design is a limitation of this research.

## Figures and Tables

**Figure 1 pathogens-11-00557-f001:**
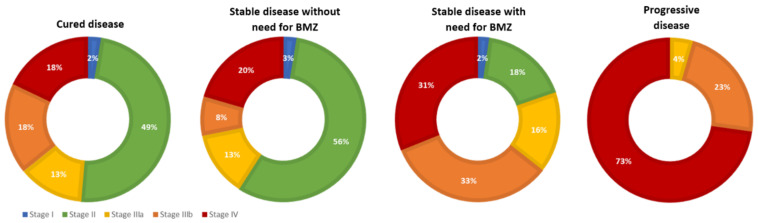
Staging according to the PNM classification across the different clinical groups.

**Figure 2 pathogens-11-00557-f002:**
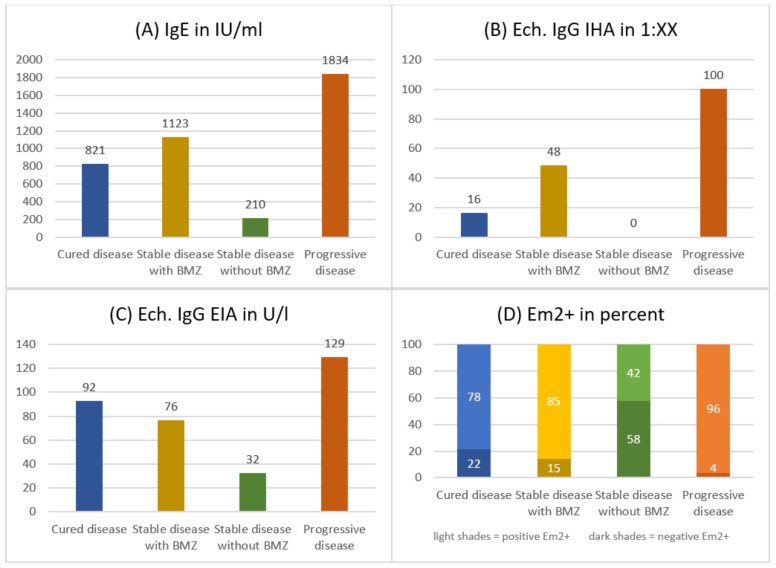
Difference in initial mean levels of IgE (**A**) and serological markers (**B**–**D**) between clinical groups.

**Figure 3 pathogens-11-00557-f003:**
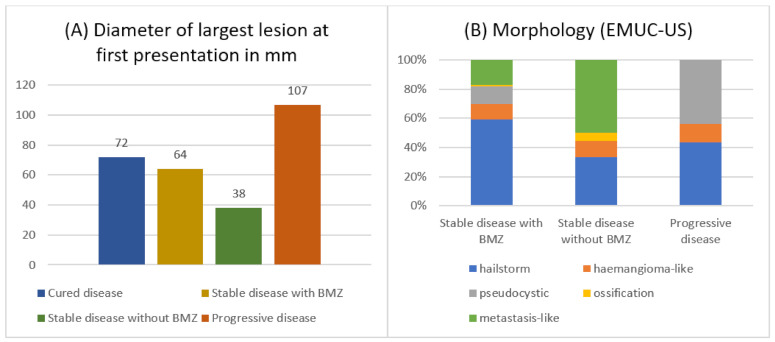
Size at first presentation (**A**) and morphology of AE lesions according to EMUC-US (**B**) across different clinical groups.

**Table 1 pathogens-11-00557-t001:** Demographic characteristics and descriptive results.

Variable	Result
**Gender**	Female 57.5%, Male 42.5%
**Mean age**	53 years (SD = 17.9 years, 11–89 years)
**Symptoms**	No symptoms 32.3%, abdominal discomfort 48.8%, abdominal pain 40.3%, loss of weight 16.7%, fatigue 15.6%, jaundice 10.5%, fever 6.2%
**WHO definition** [13] *****	possible 3.6%, probable 36.4%, confirmed 59.6%
**PNM classification** [14]	P: P0 0.4%, P1 2.9%, P2 33.1%, P3 20.1%, P4 42.7%, Px 0.8%
	N: N0 60.9%, N1 33.3%, Nx 5.8%
	M: M0 87.1%, M1 9.3%, Mx 3.6%
**Disease stage** [14]	stage I 2.8%, stage II 26.3%, stage IIIa 14.7%, stage IIIb 23.9%, stage IV 30.9%
**Outcome**	cured 14.4%, stable disease 62.2% (47.8% with BMZ, 14.4% without BMZ for M = 45.6 months, SD = 25.8), progressive disease 5.2%

* Possible case: any patient with clinical and epidemiological history and imaging findings or serology positive for AE. Probable case: any patient with clinical and epidemiological history, and imaging findings and serology positive for AE with two tests (indirect hemagglutinin, IgG EIA, Em2+ IgG, reEm18 IgG, and Echinococcus Western Blot IgG Essay). Confirmed case: (1) additional histopathology compatible with AE, and/or (2) detection of *E. multilocularis* nucleic acid sequence(s) in a clinical specimen.

**Table 2 pathogens-11-00557-t002:** Correlations between initial mean levels of IgE, serological markers, and size of the largest AE lesion.

	IgE	Ech. IgG IHA	Ech. IgG EIA	Em2+
Ech. IgG IHA	*s* = 0.412; *p* < 0.001			
Ech. IgG EIA	*r* = 0.211; *p* = 0.030	*s* = 0.542; *p* < 0.001		
Em2+	*s* = 0.442; *p* < 0.001	*s* = 0.472; *p* < 0.001	*s* = 0.419; *p* < 0.001	
Size of largest lesion	*r* = 0.438; *p* < 0.001	*s* = 0.489; *p* < 0.001	*r* = 0.431; *p* < 0.001	*s* = 0.551; *p* < 0.001

**Table 3 pathogens-11-00557-t003:** Logistic regression models for (A) progressive disease and (B) controlled disease.

Variable	(A) Progressive Disease			(B) Controlled Disease		
	OR	CI 95%	*p* =	OR	CI 95%	*p* =
Age	0.992	0.964–1.021	0.581	1.003	0.973–1.034	0.831
Gender	0.670	0.245–1.824	0.433	0.543	0.180–1.633	0.277
P—Localisation of parasite	1.886	0.824–4.313	0.133	0.668	0.335–1.330	0.251
N—Involvement of neighbouring organs	**3.696**	1.173–11.653	**0.026**	0.616	0.113–3.347	0.575
Staging (I–IV) *	**2.860**	1.384–5.911	**0.005**	0.670	0.389–1.154	0.149
Largest diameter of lesion	**1.017**	1.004–1.029	**0.007**	**0.972**	0.949–0.996	**0.022**
Morphology of lesion	0.979	0.848–1.132	0.778	1.029	0.933–1.134	0.571
IgE levels at first presentation	1.000	1.000–1.000	0.822	1.000	0.998–1.001	0.594
Echinococcus IgG EIA	1.007	0.998–1.015	0.131	0.997	0.979–1.014	0.704
Echinococcus Em2+ ELISA	1.611	0.174–14.925	0.675	**0.245**	0.072–0.835	**0.024**
γGT at first presentation	1.000	0.998–1.002	0.733	0.998	0.992–1.004	0.587

OR = odds ratio, CI 95% = 95% confidence interval, *p* = significance level with significant results printed bold. * If staging was included in the model, ‘P’ and ‘N’ were excluded. Size remained significant with *p* = 0.008.

**Table 4 pathogens-11-00557-t004:** Definition of clinical outcomes.

Clinical Outcome	Definition
**Cured** (at the end of follow-up)	Curative surgery performed >2 years ago, no signs of recurrent or residual AE lesions at study endpoint
**Stable disease with need for BMZ** (at the end of follow-up)	No curative surgery possible/wanted and stable disease with long-term BMZ treatment, i.e., no growing lesions and no occurrence of AE-associated complications; Curative surgery performed <2 years ago, ongoing adjuvant BMZ treatment.
**Stable disease without need for BMZ** (at the end of follow-up)	Controlled disease: no curative surgery possible/wanted and stable disease without BMZ treatment for at least 6 months, i.e., no growing lesions and no occurrence of AE-associated complications; Curative surgery performed <2 years ago without adjuvant BMZ treatment for at least 6 months, no signs of recurrent or residual echinococcal lesions at study endpoint.
**Progressive disease** (at any point in time during follow-up)	Growing AE-lesions in (PET-)CT-/MRI-scan; Appearance of metastases or new infiltration of neighbouring organs; AE-associated biliary/vascular complication which was newly occurring or recurring after >3 years of stable disease (complications pre-sent at first presentation did not count; patients with frequently reoccurring complications were considered ‘stable’); Recurrence after complete resection; AE-associated death.

## Data Availability

Not applicable.

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
