# Peer review of "Initial Risk Assessment in Patients with Alveolar Echinococcosis—Results from a Retrospective Cohort Study"

_pathogens, 2022, doi:10.3390/pathogens11050557_

Round 1
Reviewer 1 Report
The manuscript describes the assessment of early risk factors in patients with alveolar echinococcosis (AE). The study is valuable because the findings may contribute to better care for the patients suffering from this disease. Therefore, it is expected that this work will be particularly useful to clinicians. The manuscript is very well prepared and needs only minor corrections.
The following points should be corrected:
1. The introduction should provide background on pathogenesis, symptoms, and treatment of AE. Please expand the introduction with these issues.
2. The numbers and legend in Figure 1 are poorly visible. Please enlarge their size.
3. Line 265: Please specify which test was used to check the normality of the distribution.
4. Lines 331-332: Please correct the writing of this reference (probably a reference manager error).
Author Response
Dear sir or madam,
Thank you very much for your careful revision of our manuscript and the valuable input.
We elaborated further on the introduction and added a background regarding the disease characteristics and its treatment (line 28-42). Also, we have enlarged the figure as recommended.
Thank you for pointing out that we did not mention, which test for normality of the distribution was performed. We used the Kolmogorov-Smirnov-test or Shapiro-Wilk- and added this information to the manuscript respectively (line 287-288). We also edited the respective reference.
We hope that with the improvements made based on your recommendations, you will now find the manuscript considerable for publication. Kindly let us know, if you have any further questions or remarks.
Yours sincerely,
Lynn Peters
Reviewer 2 Report
Authors of the paper “Early risk assessment in patients with alveolar echinococcosis – results from a retrospective cohort study” described a retrospective cohort of patients with alveolar echinococcosis. They rightly notice that it is a potentially lethal parasitosis. The studies in humans’ populations are warranted because there is no control over the spread of this zoonosis and over the animal reservoir.
I must underline a serious methodological problem which limits the value of observations. The authors planned to assess early risk factors for progressive disease but in my opinion it has appeared not possible because the majority of patients was diagnosed in a late stage of disease with an involvement of neighbouring organs in 33% of patients.
First authors should define the term “progressive disease” or change the title and aims of the study. In my opinion the term “progressive disease” cannot by limited only to the observation after establishing a diagnosis. In fact, patients with an advanced disease are not comparable to those with small lesions in the beginning of observations.
I wonder what criteria have been used to diagnose alveolar echinococcosis especially in patients with negative Em2+. The diagnostic criteria for echinococcosis should be shortly described.
Authors should clearly, more precisely present data on follow up time in different groups. I have found only one, generalized piece of information about “11,228 patient months and a median of 8 visits”.
For readers it is difficult to find methodology of this clinical study in the last part of the text. This makes it difficult to interpret the results - please change it and put this section before ‘results’.
Author Response
Dear sir or madam,
Thank you very much for your careful revision of our manuscript and the valuable input.
We are sorry to have caused a misunderstanding regarding the terminology used in this manuscript: With ‘early risk assessment’ we did not mean during ‘early stage of disease’ but ‘at first presentation’. This refers to the first time a patient presents at the centre before treatment is started. We therefore thank the reviewer for pointing out this potential pitfall and changed the title and aim to ‘initial risk assessment’, as well as clarified the meaning of the term used the manuscript (line 49-50).
The outcome ‘progressive disease’ is defined in table 4 and includes all patients with:
- growing AE-lesions in (PET-)CT-/MRI-scan
- appearance of metastases or new infiltration of neighbouring organs
- AE-associated biliary/vascular complication which was newly occurring or recurring after > 3 years of stable disease (complications present at first presentation did not count; patients with frequently reoccurring complications were considered ‘stable’)
- recurrence after complete resection
- AE-associated death
According to the journal’s structure, the methods are being placed at the end of the manuscript; however, in this paper it might enhance understanding if the methods could be placed between the introduction and the results and would kindly ask the editors to consider this proposal.
Regarding the definition of ‘progressive disease’, we would like to emphasize that this category is not to be confused with ‘extensive disease’ which is used widely in the literature regarding AE. Untreated AE can result in extensive disease, meaning the involvement of neighbouring organs and distant metastasis. Still, patients with extensive disease can respond to treatment, e.g. with shrinking lesions. On the other hand, patients with ‘average sized’ AE of the liver can progress in spite of treatment. Other patients do not seem to need treatment at all. We assume that underlying immunological and genetic mechanisms are responsible for the course of disease and the response to treatment. However, our aim was to assess early warning signs at initial presentation that might indicate the course of disease and support the clinicians working with AE patients.
To establish the diagnosis of AE, the definition from Brunetti et al. (2010) was applied (c.f. table 1). We included the exact definition in the manuscript (c.f. legend of table 1) to clarify our approach. Overall, 3.6% had possible AE, 36.4%, probable AE and 59.6% confirmed AE. In patients that present with a hepatic lesion and only one positive serological test result available at our centre (including indirect hemagglutinin, IgG EIA, Em2+ ELISA), a biopsy is frequently taken and/or tests from external laboratories (e.g. the German reference centre in Würzburg) are used, mostly a western blot or the reEm18-ELISA to establish the diagnosis.
We added additional information regarding the follow-up of our cohort and the respective groups (line 54-56).
We hope that with the improvements made based on your recommendations, you will now find the manuscript considerable for publication. Kindly let us know, if you have any further questions or remarks.
Yours sincerely,
Lynn Peters
Reviewer 3 Report
Dear sir,
thank you to select me to review manuscript: Peters L et al. Early risk assessment in patients with alveolar echinococcosis - results from a retrospective cohort study. The authors included 279 patients with alveolar echinococcosis into final analysis, the study has retrospective design. The authors concluded that a large AE lesion, PNM staging and especially the involvement of neighbouring organs were significant risk factors for progressive disease in the multivariate analysis. Study is well made, results are clearly presented, statistical analysis is proportional, discussion is adequate and conclusions are clear. Only minor changes are needed:
Please add decision of local Ethic committee.
Abstract, lines 15-19: please add confidental intervals
Results, lines 75-80: Please add liver enzyme test levels and conclude, that between all groups wasn´t found statistical significance
Multivariate analysis, line 116: please data about adjustation
Limitations, lines 296-300: retrospective and monocentric design of the study were major limitations.
Author Response
Dear sir or madam,
Thank you very much for your careful revision of our manuscript and the valuable input, as well as the overall positive feedback. According to your suggestions, we included the decision of the Ethical board in the methods section (line 232), as well as the confidence intervals to the abstract (line 16-19).
Regarding the level of liver enzymes across the respective groups, these values are listed in the supplementary material S1 along with the exact statistical metrics and test results. In the main body of the manuscript, we aimed to focus on the most important results. However, we added the reference to the supplementary material to the text and hope that you can agree with us that this will enhance clarity without comprising the data.
We were uncertain what is meant by the comment ‘please data about adjustion’. We used multivariate analysis as a method to control the mutual effect of the variables and effect modifiers (sex, age) on the outcome. Hence, we used the term ‘adjusting’.
We included the mentioned limitations, thank you very much for this remark (line 324-325).
We hope that with the improvements made based on your recommendations, you will now find the manuscript considerable for publication. Kindly let us know, if you have any further questions or remarks.
Yours sincerely,
Lynn Peters
Round 2
Reviewer 2 Report
Thank you for the response. I accept the paper in the present form with one liitle exception:
Line 95 – for liver function enzymes the term “activity” should be used instead “level”. Please correct it.
Author Response
Dear sir or madam,
thank you very much for your revision. We will make the recommended change.
Kind regards,
Lynn Peters